# Passive Drag in Young Swimmers: Effects of Body Composition, Morphology and Gliding Position

**DOI:** 10.3390/ijerph17062002

**Published:** 2020-03-18

**Authors:** Matteo Cortesi, Giorgio Gatta, Giovanni Michielon, Rocco Di Michele, Sandro Bartolomei, Raffaele Scurati

**Affiliations:** 1Department for Life Quality Studies, University of Bologna, 40132 Bologna, Italy; giorgio.gatta@unibo.it; 2Department of Biomedical Sciences for Health, Università degli Studi di Milano, 20129 Milan, Italy; giovanni.michielon@unimi.it (G.M.); raffaele.scurati@unimi.it (R.S.); 3Department of Biomedical and Neuromotor Sciences, University of Bologna, 40132 Bologna, Italy; rocco.dimichele@unibo.it (R.D.M.); sandro.bartolomei@unibo.it (S.B.)

**Keywords:** gliding, performance, streamline, swimming, young athletes, resistive forces, exercise testing, body composition

## Abstract

The passive drag (Dp) during swimming is affected by the swimmer’s morphology, body density and body position. We evaluated the relative contribution of morphology, body composition, and body position adjustments in the prediction of a swimmer’s Dp. This observational study examined a sample of 60 competitive swimmers (31 male and 29 female) with a mean (±SD) age of 15.4 ± 3.1 years. The swimmer’s Dp was measured using an electro-mechanical towing device and the body composition was assessed using a bioelectrical impedance analyser. Body lengths and circumferences were measured in both the standing position and the simulated streamlined position. Partial correlation analysis with age as a control variable showed that Dp was largely correlated (*p* < 0.05) with body mass, biacromial- and bi-iliac-breadth, streamline chest circumference and breadth. Body mass, Body Mass Index, chest circumference and streamline chest circumference showed a significant and moderate to strong effect (η2 > 0.55) on Dp. Body mass was the best predictor of Dp explaining 69% of the variability. These results indicate that swimmers with lower Dp values were: (i) slimmer, with lower fat and fat-free mass, (ii) thinner, with lower shoulder breadth, chest circumference, and streamline trunk diameters (iii), shorter, with lower streamline height. These findings can be used for talent identification in swimming, with particular reference to the gliding performance.

## 1. Introduction

Swimming velocity results from the interaction between the swimmer’s propulsion and drag factors. The swimmer’s drag increases at least quadratically as velocity raises, thereby affecting the energy expenditure. Hence, swimmers take great advantage by reducing, as much as possible, the drag forces, either in passive (e.g., during gliding, Dp) and active (e.g., while swimming) conditions [1].

Several studies considered the effects of the body structure and shape on the swimmer’s Dp, relating morphological indices such as height, arm span, chest, appendicular segmental circumferences, and body breadths to glide efficiency [2]. Close correlations were shown between Dp and body structure and shape, particularly body mass, height and trunk diameters [3,4,5]. While controversial issues exist due to the difficulty to accurately measure the wetted surface area of the swimmer’s body [3,6], the literature largely agrees that the cross-sectional area, the length of the body and the body mass are the main parameters involved in determining the swimming Dp [3,7,8,9]. 

The swimmer’s body position in water is affected by the buoyancy of body masses. During the streamline attitude, a typical position of swimming gliding, underwater torque force acts on the body and causes the legs to sink [10]. The body density has been demonstrated to affect the body torque and the literature agrees to assert that individual body density and morphology can impact the Dp [11]. Further measurements made under dynamic conditions showed that higher body density induces a trunk incline, increasing the drag [12,13]. Thus, it is generally acknowledged that swimmers with higher fat percentage have better body buoyancy [11,14].

The adjustments performed by the swimmer during gliding may be viewed as technical characteristics which can reduce the passive drag [15]. Since a direct relationship exists between a swimmer’s cross-sectional area and his/her drag, to reduce the amount of drag while gliding, swimmers must avoid, as far as possible, any movement altering the body alignment [16]. In particular, the position of the head and shoulders [2,17,18], breathing maneuvers [19], lumbar alignment [20] and torso shapes [21] can be modulated to get an effective streamline position, possibly acting on the swimmer’s shape itself.

Several authors examined the combined effects of morphological and body composition characteristics on Dp [9,22], but studies are still lacking that assess which among the characteristics described above, including a swimmer’s technical adjustments, are related to the gliding position. Therefore, the study aimed to investigate the impact of morphological characteristics, body composition, and technical characteristics on the Dp in young competitive swimmers. Since Dp is a multifactorial measure which is affected by different variables, we evaluated the relative contribution of morphology, body composition, and technical characteristics in the prediction of a swimmer’s Dp. 

## 2. Materials and Methods 

### 2.1. Participants

Sixty young regional swimmers (29 females and 31 males) participated in this study. The participants’ characteristics are shown in Table 1. The following inclusion criteria were adopted: (i) a minimum of 5 years of previous swimming competitive experience; (ii) chronological age between 12 and 19 years old; (iii) swimmer able to compete in all the Qlympic swimming strokes and distance; (iv) swimmer that participated in the training for a minimum of 4 sessions per week in the last 12 weeks. Data were collected during the spring, when the swimmers were in their competition period. All participants were non-smokers, and none of them was following specific dietary interventions. The participants did not perform any other physical activity in the day of data collection. All swimmers received written and oral instructions before the study and gave their written informed consent to the experimental procedure. The experimental protocol was approved by the Ethics Committee of the University of Milan (Approval nr. 35/18, 17 July 2018).

### 2.2. Design and Methodology

The study was designed as an observational study with one data collection session for any participant. Data were collected in a 25 m indoor swimming pool (average water temperature: 28.0 ± 0.5 °C) in the morning time (9:00–12:00). Each swimmer spent an average of 2 h at the pool. The study protocol was divided into three parts: (i) on-land assessments: body composition, morphological and technical characteristics were collected; (ii) familiarisation with the in-water assessments: before Dp data collection, swimmers performed three towing trials to become familiar with the assessments; (iii) in-water assessments: a passive towing protocol was performed. To preserve blinding, the participants were not informed about their individual scores until the end of the study. The study-related characteristics were completely random and none of the participants dropped out of the study.

### 2.3. On-land Assessments

Body composition was measured using an 8-contact electrode impedance analyser system (body composition analyser BC-418AM, Tanita Corporation, Tokyo, Japan) capable of acquiring segmental body composition analysis [23]. The 8-electrode bioelectrical impedance analysis (BIA) method can provide an accurate measurement of segmental and total body compositions in athletes [24]. The fat (FM) and fat-free mass (FFM) of arms, trunk and legs were the body composition variables examined in this study. During the measurement, the subjects stood on the platform of the BIA device with both feet in contact with the electrode, while holding the hand grips with the electrodes. A low voltage current passed through the body. BIA measurements were carried out at a frequency of 50 kHz (550 mA) and the swimmers were simultaneously weighed (kg). Participants were asked not to drink any beverage, starting from two hours before the measurement procedure. The BIA measurement was repeated three times for each athlete, and the mean value was used for further analyses. All measurements were performed by a single investigator and were carried out in a temperature- and humidity-controlled room. Each participant completed the BIA protocol within ten minutes.

Concerning morphological parameters, height, arm span, biacromial-, bideltoid- and bi-iliac- breadths and the chest circumference at the level of mid sternum were measured in a standing position. Furthermore, the body mass index (BMI) was estimated as weight in kilograms divided by the square of height in meters.

Body height, biacromial breadth, chest breadth, chest depth and chest circumference in the streamline position were selected as technical parameters because the cross-sectional area and the length of the body were considered as the main body size influencers of Dp performance [7,9]. These technical variables were measured on land in a standing position, simulating the streamlined position adopted during the underwater glide as closely as possible.

Body height was measured in the upright position from the vertex to the floor and from akropodion to dactylion (fingertips to tiptoes) for morphological and technical parameters, respectively. Biacromial breadths were always considered as the distance between the two acromion processes. Bideltoid breadth was the width of the shoulders at the widest point (clearance at shoulder level). Chest depth was measured from the back and the highest point of the chest at the level of the xiphoid process of the sternum. Chest breadth was the breadth of the torso at nipple level. Bi-iliac breadth was measured as the distance between the left and right anterior superior iliac spines.

All on-land measurements were obtained after a maximum inhalation and wearing only a dry textile swimsuit. Body lengths were measured using a large anthropometer (Metrica 10455, San Donato Milanese, Italy), while circumferences were measured with a non-elastic tape at the level of mid sternum.

### 2.4. In-water Assessments

The in-water session included ten 25 m towing programmed trials of Dp measurement (at a constant towing speed of 0.8, 1.0, 1.2, 1.4, 1.6, 1.8, 2.0, 2.2, 2.4, 2.6 m/s), which were separated by 3-minute pauses. The Dp of swimmers was measured using an electro-mechanical device (Swim-Spektro, Talamonti Spa, Ascoli Piceno, Italy). Swim-Spektro consists of a low-voltage isokinetic engine. It was placed at the edge of the pool and towed a swimmer along the pool via a non-elastic wire. The measurement procedure was performed by asking the swimmers to assume the best hydrodynamic glide position (streamline) during the whole duration of towing, at the waterline. The swimmers were required to hold the lower limbs and feet at maximum extension, the upper limbs extended at the elbow, and to position the upper arms in contact with the sides of the head with the wrist joints, one hand on top of the other. Participants were also asked to hold their breath after a maximal inhalation during the whole towing. The streamline position of the swimmers was carefully checked through visual observation by two operators. Technical suggestions during the trials were deliberately avoided to not affect the behaviour of participants while being towed.

Towing force data were captured from the towing system and recorded using a dedicated software (DB:4, Talamonti Spa, Ascoli Piceno, Italy). The Dp exerted by the swimmer was equal to the instantaneous force required for towing. For the current analysis, we considered only the data acquired from 15 to 5 m from the device, when the speed was constant. Dp values of each trial (Dp = k **•** v^2^, N) were divided by the square of the corresponding speed (v^2^, m^2^
**•** s^−2^) to obtain the speed-specific drag (k coefficient = Dp/v^2^, N **•** m^−2^
**•** s^−2^). Towing velocity and towing force were calibrated at the beginning of each in-water session, as shown in a previous study [25]. To quantify the reliability of the towing methods, ten participants repeated the Dp measurements five times for each speed.

### 2.5. Statistics

The normality and homoscedasticity assumptions were assessed on all data using Kolmogorov–Smirnov and Levene tests, respectively. All data met these assumptions. Means and standard deviations were calculated for all parameters. Partial correlation coefficients (Rp) with age as a control variable were used to determine the degree of association between assessment variables and Dp. In addition to statistical significance, the following criteria were adopted for interpreting the magnitude of correlation (r): <0.1, trivial; 0.1–0.3, small; 0.3–0.5, moderate; 0.5–0.7, large; 0.7–0.9, very large; and 0.9–1.0, almost perfect.

The total sample was divided into quartiles according to the values of the K coefficient. Group means were compared using a two-way ANOVA, with sex and K level as factors, and post-hoc Bonferroni tests were used to verify differences between quartiles (*p* < 0.05). The effect size based on the eta-squared (η^2^) was computed, and interpreted as (i) without effect if 0 < η^2^ < 0.04; (ii) minimum if 0.04 < η^2^ < 0.25; (iii) moderate if 0.25 < η^2^ < 0.64 and; (iiii) strong if η^2^ > 0.64 (Ferguson, 2009). 

Stepwise linear regression analyses were used to assess potential relationships of the parameter’s variables (i.e., body composition, morphological and technique) with Dp (K coefficient) and to evaluate which parameters best characterized Dp. Additionally, a multiple linear regression model using a stepwise procedure was repeated, entering all variables. A *p*-value threshold of 0.05 was used for statistical significance. SPSS for Windows, version 20.0 (SPSS Inc.; Chicago, IL, USA) was used for all analyses.

## 3. Results

Repeatability results showed high intra-subject agreement for Dp: ICC was always higher than or equal to 0.93. Descriptive statistics for body composition, morphological and technique parameters and their relationship with Dp, with age as control variable, are presented in Table 2. Partial correlation analysis showed that Dp was largely correlated (*p* < 0.05) with body mass, BMI, biacromial- bi-iliac-breadth, streamline biacromial- chest-breadth, streamline chest circumference, total- legs- trunk-FFM and total body water (see Table 3). All other correlations were at least moderate.

Table 2 and Table 3 also show the effects of Dp level and sex. Drag performance was higher in quartile 1 and decreased up to quartile 4 with a significant and strong effect. The body mass, BMI, chest circumference and streamline chest circumference showed a significant and almost strong effect (η^2^ > 0.55) on Dp level. A significant and moderate effect for all other variables was observed (Table 2). 

Sex had a significant and moderate effect on arm span, chest circumference, streamline chest depth and all body composition variables with the exception of arms FM. There was a non-significant effect of sex only regarding BMI and bi-iliac breadth. The interaction between sex and Dp level was significant only for FM and legs arms- trunk-FM with moderate effects, for legs and arms FM with minimum effect.

The outcome of the multiple linear regression analysis performed with all examined variables as predictors of Dp are presented in Table 4a, Table 4b, Table 4c and Table 4d, respectively. Using regression analysis modeling, the best predictor of Dp performance among body composition variables was trunk FFM, while it was body mass among morphological variables and streamline chest circumference among technique variables. The single components explained 56%, 69% and 65% of variance in Dp, respectively, for body composition, morphological and technical characteristics. Body mass was the best predictor of Dp, explaining 69% of variability, as shown by the multiple linear regression comprising all variables.

## 4. Discussion

The aim of this study was to investigate the impact of morphological features, body composition, and technical characteristics on Dp in young swimmers. The literature previously reported that Dp is affected by morphological features and technical characteristics of swimmers, but the contribution of these features to determine Dp had not yet been highlighted. When comparing only body composition and anthropometric variables, body mass was found to be the factor most affecting Dp [9,22]. The main findings of this study were (1) that a similar level of Dp prediction was shown by the most important predictor within any group of examined parameters (56.3%, 68.7%, and 65.5% of explained Dp variance for body composition, morphology and technique parameters, respectively); (2) that body mass was the best overall predictor of Dp performance in youth swimmers, explaining 68.7% of variance; (3) that body composition, morphological and technique characteristics were moderately to very largely correlated to swimmer’s drag performance; and (4) that there was a moderate effect of sex on Dp. 

In this study, Dp was used to divide swimmers into quartiles: the first quartile represents the group with the best (lower) Dp, while the fourth quartile includes the swimmers with the worst (higher) Dp. Swimmers in the first quartile, in comparison with those in other quartiles, were: (i) slimmer, with lower FM and FFM, (ii) thinner, with lower shoulder breadth, chest circumference, and streamline trunk diameters (iii), shorter, with lower streamline height. 

Dp can be considered as significantly contributing to the prediction of gliding performance in swimming [4]. Previous studies in this field have found, on average, a K coefficient close to 25 for adult swimmers and lower drag coefficient for young swimmers [15]. The lower Dp in youth may be affected by the body shape [26]. Conversely, the way that young swimmers opposed when passively towing in the hydrodynamic position seems not to be affected by sex, in agreement with other studies conducted in 13–14 years-old swimmers [9]. The values of the K coefficient observed here, ranging from a mean of 16.3 in the first quartile to 25.1 in the fourth quartile, are consistent with the previous literature. The absence of differences between sexes in young swimmers can be explained by the low variability of morphological and body composition parameters as compared to adulthood [27].

### 4.1. Passive Drag and Morphology

Correlations were shown in this study between Dp and selected morphological parameters. These findings are consistent with previous reports [3,4,22], showing that Dp of young male and female swimmers strongly correlates with body mass, height and trunk diameters. In the present study, greater body mass, greater biacromial/bi-iliac/bideltoid breadth and greater height and chest circumference were all associated to greater Dp, supported by greater cross-sectional area. It is worth noting that a close relationship was confirmed between body size and Dp [26]. This relationship may suggest the importance of body size to determine Dp. Indeed, irregularity in the body curvature creates a turbulent flow and increases form drag. Tapered body shape should be designed to minimize the drag due to the turbulent flow around the body, which implies that the body shape should be somewhat “egg-shaped” [2]. These characteristics can explain the correlation between trunk sizes and Dp observed here.

Furthermore, Dp was better predicted by body mass in comparison to other morphological parameters. This finding is consistent with findings of Benjanuvatra et al., [22] and Lyttle et al., [9] that showed that body mass is the main factor contributing to increased Dp when all morphological and body composition parameters are analysed together. Larger swimmers have higher drag compared to smaller swimmers, and everything else being equal, the main contribution to Dp is due to body mass. In conclusion, anthropometric indices can provide a better indication of how body morphology affects Dp.

### 4.2. Passive Drag and Body Composition

The hydrodynamic profile of the swimmer is considered a multifactorial phenomenon and different variables contribute to the passive floating and gliding. In this study, only few body composition parameters (in particular, FFM) were shown to be related to Dp performance, and trunk FFM was the best predictor of performance among body composition variables. 

Few studies have investigated the relationship between Dp and body composition parameters. Nevertheless, body composition is considered a key factor affecting buoyancy and thus, the resistive force. It is generally acknowledged that swimmers with higher fat percentage have higher net buoyancy. Furthermore, a different distribution of body tissues can affect the buoyancy and the torque around the body mass center [11,14]. In particular, greater FFM tends to cause the body to sink. According to previous research [28], the swimmer’s drag and, more specifically, the contribution of the wave drag to the total drag force, increases with the depth of the swimmer. The body sinking may decrease the wave drag contribution and consequently, the Dp, which can explain the positive correlation between the FFM and the K coefficient, in agreement with a previous study [29]. Thus, the Dp difference between swimming bodies depends also on the differences in the body composition characteristics, as well as on morphological indices.

### 4.3. Passive Drag and Gliding Technique

Adjustments of gliding position are used by the swimmer according to the best streamline human’s profile. A group of researchers suggest that modifications in the body–fluid interface influence hydrodynamic resistance [30]. The streamline position adopted during passive gliding becomes increasingly important to minimize the drag: the reduction of the swimmer’s frontal area and the non-uniformity of the body shape are key factors for performance, particularly during the start and the turns. Indeed, when the fluid hits impact points along the swimmer’s body (i.e., head, shoulders, glutei), the local pressure increases and the fluid speed decreases, resulting in the generation of vortexes and an increase in turbulence and pressure drag [31]. The characteristics of the streamline profile are not fixed values, but they vary with technical adjustments: the swimmer can stretch the body to reach a longer length shape or correct his/her trunk segment configuration by adopting maneuvers to reduce Dp [28]. Maruyama and Yanai [19] confirmed that a significant decrease of Dp occurred after abdominal maximal inspiration in comparison to chest maximal inspiration, due to a better streamline shape configuration. In this study, a larger chest section in the streamline position was correlated with higher Dp. These results are consistent with a previous work [3], and the findings could be explained by the reduction of the depths of concavity and convexity of the trunk segment. Furthermore, Dp was better predicted by the streamline chest circumference in comparison to other technical features among those examined here. As suggested by Chatard et al. [4], the thinner the trunk is kept, the greater a swimmer’s advantage due to reduced resistive forces would be. In conclusion, by altering the body shape using technical adjustments, particularly for streamline chest circumference, a swimmer can reduce his/her Dp.

### 4.4. Practical Applications, Limitations and Future Perspectives

The main limitations of this research may be summarized as follows: (i) regarding the technical characteristics, not all the characteristics of the swimmer’s body alignment were considered in the study (e.g., changes in head position or knee and hip joint angles can also be considered as technical adjustments); (ii) regarding the methodological approach, despite the good reliability of the method, the flexibility of the human body anatomy does not guarantee a stable alignment in gliding as in a mannequin. Furthermore, although most of the previous research assessed Dp in youth swimmers even with smaller sample sizes than the present study, research with adult/elite swimmers will be needed to investigate the best predictors of Dp in those swimmers. 

The findings of the current study can have important implications both for coaches and swimmers. Coaches can consider body shape characteristics as a main factor when trying to improve a swimmer’s gliding. For example, swimmers with lower streamline height and lower streamline trunk diameters are able to glide more efficiently compared with swimmers with higher streamline height and higher streamline trunk diameters. Moreover, to favorably change their Dp, swimmers should keep a thin chest shape by using technical adjustments. It should be pointed out that adopting a shape with active maneuvers like breathing should improve glide performance associated to varying body characteristics.

Further studies are warranted to investigate other morphological aspects that may be relevant to swimming performance. Furthermore, other technical adjustments performed by a swimmer during gliding in passive streamline such as lumbar and cervical alignments can be explored in future investigations.

## 5. Conclusions

The findings of this study indicate that swimmers with lower Dp are slimmer, thinner, and shorter in comparison with swimmers with high Dp. Our results support the fact that the body mass is the main parameter involved in determining the swimming Dp, despite the body composition, the morphological and the technique characteristics were all largely correlated to the swimmer’s drag performance. Furthermore, the Dp performance seems unaffected by sex in the young swimmer. The coaches can consider body shape characteristics as the main factor when trying to improve a swimmer’s gliding and these findings can be used for talent identification in gliding performance.

## Figures and Tables

**Table 1 ijerph-17-02002-t001:** Participants demographics. Data are reported as mean (±SD).

	Age (yrs)	Height (cm)	Body Mass (kg)	Best Time 50 m Front Crawl (Long Course, s)	Level (FINA Points Long Course)	Swimming Experience (yrs)	Weekly Swim Sessions (n/Week)
Females (n = 29)	15.6 (±3.4)	163.2 (±7.3)	55.0 (±8.9)	30.91 (±1.70)	462.9 (±81.3)	6.4 (±1.1)	6 (±1)
Males (n = 31)	15.2 (±2.9)	172.0 (±11.5)	62.6 (±11.8)	26.96 (±2.04)	481.5 (±107.3)	6.5 (±1.3)	6 (±1)
All (n = 60)	15.4 (±3.1)	167.7 (±10.7)	59.9 (±11.3)	28.98 (±2.72)	471.7 (±93.8)	6.5 (±1.2)	6 (±1)

**Table 2 ijerph-17-02002-t002:** Descriptive statistics according to best performance in K coefficient.

Variable	Total (n = 60)	Quartile 1 (n = 15)	Quartile 2 (n = 15)	Quartile 3 (n = 15)	Quartile 4 (n = 15)
K Coefficient (N)	20.65 (±3.44)	16.3 (±1.66)^2,3,4^	19.52 (±0.72)^1,3,4^	21.72 (±0.81)^1,2,4^	25.06 (±1.42)^1,2,3^
Body Composition characteristics					
FM (kg)	11.44 (±3.76)	9.02 (±2.47)^2,3,4^	12.07 (±2.86)^1,4^	11.03 (±2.29)^1,4^	13.63 (±5.29)^1,2,3^
FFM (kg)	46.83 (±10.64)	36.63 (±7.1)^2,3,4^	43.17 (±6.41)^1,4^	50.93 (±6.69)^1,4^	56.61 (±9.72)^1,2,3^
Legs FM (kg)	5.21 (±1.71)	4.27 (±1.21)^2,3,4^	5.65 (±1.29)^1,4^	5.03 (±1.35)^1,4^	5.87 (±2.39)^1,2,3^
Legs FFM (kg)	15.54 (±3.89)	11.89 (±2.68)^2,3,4^	14.3 (±2.87)^1,4^	16.94 (±2.62)^1,4^	19.01 (±3.26)^1,2,3^
Arms FM (kg)	1.48 (±0.38)	1.24 (±0.25)^2,4^	1.52 (±0.33)^1,4^	1.43 (±0.26)^4^	1.72 (±0.49)^1,2,3^
Arms FFM (kg)	4.53 (±1.48)	3.26 (±0.89)^2,3,4^	3.93 (±0.87)^1,4^	5.03 (±0.9)^1,4^	5.89 (±1.56)^1,2,3^
Trunk FM (kg)	4.78 (±1.91)	3.52 (±1.24)^4^	4.92 (±1.52)^4^	4.59 (±0.93)^4^	6.07 (±2.69)^1,2,3^
Trunk FFM (kg)	26.8 (±5.41)	21.45 (±3.68)^2,3,4^	24.99 (±2.83)^1,4^	28.99 (±3.36)^1,4^	31.75 (±5.02)^1,2,3^
Morphological characteristics					
Height (cm)	167.15 (±11.13)	156.67 (±10.42)^2,3,4^	166.2 (±8.2)^1^	172.27 (±7.74)^1^	173.47 (±9.83)^1^
Body mass (kg)	58.27 (±11.69)	45.64 (±8.25)^2,3,4^	55.23 (±7.13)^1,4^	61.95 (±7.19)^1,4^	70.23 (±7.51)^1,2,3^
Bmi (kg/m^2^)	20.66 (±2.53)	18.44 (±1.65)^2,3,4^	19.99 (±2.1)^1,4^	20.82 (±1.33)^1,4^	23.38 (±2.07)^1,2,3^
Arm span (cm)	175.25 (±12.99)	163.3 (±10.83)^2,3,4^	172.8 (±10.12)^1^	179.9 (±8.29)^1^	185 (±11.78)^1^
Chest circum. (cm)	82.49 (±7.99)	74.23 (±5.01)^2,3,4^	79.63 (±5.21)^1,4^	85.5 (±5.6)^1,4^	90.6 (±4.81)^1,2,3^
Biacromial breadth (cm)	31.44 (±2.91)	29 (±2.09)^2,3,4^	30.8 (±1.85)^1,4^	32 (±2.05)^1^	33.97 (±3.13)^1,2^
Bideltoid Breadth (cm)	42.28 (±3.59)	38.8 (±2.26)^2,3,4^	41.27 (±2.24)^1,4^	43.17 (±2.32)^1,4^	45.87 (±3.16)^1,2,3^
Bi-iliac breadth (cm)	29.2 (±2.48)	26.87 (±1.71)^2,3,4^	28.8 (±1.92)^1,4^	29.77 (±2.14)^1,4^	31.37 (±1.85)^1,2,3^
Technical characteristics					
Streaml. height (cm)	230.93 (±15.96)	215.6 (±13.53)^2,3,4^	229.47 (±12.56)^1^	237.37 (±10.19)^1^	241.3 (±14.67)^1^
Streaml. biacromial breadth (cm)	34.74 (±3.34)	31.7 (±1.98)^2,3,4^	33.37 (±2.52)^1,4^	35.93 (±2.54)^1,4^	37.97 (±2.36)^1,2,3^
Streaml. chest breadth (cm)	29.88 (±3.22)	27.13 (±1.98)^2,3,4^	29 (±3.16)^1,4^	30.67 (±2.48)^1,4^	32.73 (±2.33)^1,2,3^
Streaml. chest depth (cm)	20.66 (±2.21)	18.87 (±1.36)^2,3,4^	19.87 (±1.73)^1,4^	21.63 (±2.22)^1^	22.27 (±1.7)^1,2^
Streaml. chest circum. (cm)	88.71 (±7.87)	80.77 (±5.18)^2,3,4^	85.87 (±6.02)^1,4^	91.9 (±4.63)^1,4^	96.3 (±5.24)^1,2,3^

Note: ^1^ post-hoc significant difference with quartile 1 obtained by ANOVA (*p* ≤ 0.05) for K coefficient, ^2^ with quartile 2, ^3^ with quartile 3, ^4^ with quartile 4; circum.: circumference; streml.: streamline.

**Table 3 ijerph-17-02002-t003:** Inferential analysis and effect size of the variables analyzed.

Variable	Partial Correlation with K coef (R*p*)	Quartile Effect (p)	Quartileeffect (η^2^)	Sex Effect (p )	Sex Effect (η^2^)	Interaction Effect (p)	Interaction Effect (η^2^)
K Coefficient (N)		<0.001	0.874	0.654	0.004	0.731	0.024
Body Composition characteristics							
FM (kg)	0.471 *	<0.001	0.449	<0.001	0.382	0.001 *	0.260
FFM (kg)	0.734 *	<0.001	0.517	<0.001	0.333	0.250	0.075
Legs FM (kg)	0.340*	<0.001	0.357	<0.001	0.467	0.028 *	0.159
Legs FFM (kg)	0.711 *	<0.001	0.503	<0.001	0.396	0.286	0.069
Arms FM (kg)	0.485 *	<0.001	0.328	0.016 *	0.107	0.045 *	0.143
Arms FFM (kg)	0.698 *	<0.001	0.465	<0.001	0.363	0.344	0.061
Trunk FM (kg)	0.527 *	<0.001	0.454	<0.001	0.251	0.000 *	0.306
Trunk FFM (kg)	0.745 *	<0.001	0.517	<0.001	0.255	0.163	0.093
Morphological characteristics							
Height (cm)	0.660 *	<0.001	0.333	0.002 *	0.177	0.218	0.081
Body mass (kg)	0.825*	<0.001	0.584	0.021 *	0.099	0.813	0.018
Bmi (kg/m^2^)	0.752 *	<0.001	0.551	0.513	0.008	0.110	0.109
Armspan (cm)	0.652 *	<0.001	0.396	<0.001	0.310	0.145	0.098
Chest circum. (cm)	0.666 *	<0.001	0.615	<0.001	0.295	0.499	0.044
Biacromial breadth (cm)	0.758 *	<0.001	0.347	0.009 *	0.124	0.212	0.082
Bideltoid Breadth (cm)	0.682 *	<0.001	0.501	0.003 *	0.156	0.372	0.058
Bi-iliac breadth (cm)	0.799 *	<0.001	0.431	0.120	0.046	0.316	0.065
Technical characteristics							
Stream. height (cm)	0.626 *	<0.001	0.375	<0.001	0.251	0.196	0.085
Streaml. biacromial breadth (cm)	0.768 *	<0.001	0.495	0.001 *	0.182	0.548	0.040
Streaml. chest breadth (cm)	0.710 *	<0.001	0.390	0.003*	0.162	0.491	0.045
Streaml. chest depth (cm)	0.626 *	<0.001	0.356	<0.001	0.301	0.215	0.082
Streaml. chest circum. (cm)	0.810 *	<0.001	0.556	<0.001	0.228	0.620	0.033

Note: Rp: partial correlation coefficients with age as control variable; p and η2: two-way analysis of variance results for quartile and sex effect. * p ≤ 0.05; circum.: circumference; streml.: streamline.

**Table 4 ijerph-17-02002-t004:** Stepwise linear regression model summary for all components.

Model	R	R^2^	Adjusted R^2^	*SE* of the Estimate	F Value
a. Body Composition					
1	0.755	0.570	0.563	2.25	79.576 *
2	0.841	0.708	0.698	1.87	27.867 *
b. Morphological					
1	0.832	0.693	0.687	1.92	132.881 *
2	0.848	0.719	0.709	1.85	5.423 *
c. Technique					
1	0.813	0.661	0.655	2.01	115.113 *
d. All Variables					
1	0.832	0.693	0.687	1.92	132.881 *
2	0.848	0.719	0.709	1.85	5.423 *

a.1 Predictors: (Constant), Trunk FMM; a.2 Predictors: (Constant), Trunk FMM, Trunk FM; b.1 Predictors: (Constant). Body mass; b.2 Predictors: (Constant). Body mass, BMI; c.1 Predictors: (Constant). Streamline chest circumference; d.1 Predictors: (Constant). Body mass; d.2 Predictors: (Constant). Body mass, BMI; Note: * *p* ≤ 0.05.

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
