# Peer review of "Passive Drag in Young Swimmers: Effects of Body Composition, Morphology and Gliding Position"

_ijerph, 2020, doi:10.3390/ijerph17062002_

Round 1
Reviewer 1 Report
I am grateful for the opportunity to review the manuscript presented to me. I believe the paper is worth considering for publication, however requires major revision.
line 19 - please explain the abbreviation BMI
line 62-69 - please specify the aim of the study
line 147 - how was the statistical test power?
line 147 - what was the distribution of variables?
line 171-174 - I think that the author should accept the consequence of writing data, since he writes the BMI abbreviation, and the fat, fat-free or total body water should be saved as an abbreviation, e.g. FM, FFM, TBW
table 2 should be in a horizontal orientation - it will be more readable
the description of table 2 should be much shorter, while the current information in the table title moved under the table as a legend. This note also applies to table 3 and 4
table 3 - I think p = 0.000 is incorrect, I suggest changing to p <0.001 everywhere
I would suggest modifying the presentation of the conclusions
You should change the style of citing from Harvard over Vancouver - non-compliance with the journal's guidelines
Author Response
I am grateful for the opportunity to review the manuscript presented to me. I believe the paper is worth considering for publication, however requires major revision.
Answer: We would like to thank the Reviewer for his/her comments that improved the quality of the manuscript. The manuscript was modified according to the suggestion of the Reviewer (see point by point answers below). We hope to have addressed all his/her concerns.
line 19 - please explain the abbreviation BMI
Answer line 19: as suggested by the Reviewer, the abbreviation "Body Mass Index" was added before the acronym “BMI”
line 62-69 - please specify the aim of the study
Answer line 62-69: we thanks the reviewer for his comment. The purpose of the study is now clearly specified in the last paragraph of the Introduction section.
line 147 - how was the statistical test power?
Answer: since in this study there are many different variables and also we performed different statistical analyses (partial correlations, ANOVAs, linear regressions), there would be a huge number of power values to be presented. Thus, for simplicity, we decided not to report power values in the study.
line 147 - what was the distribution of variables?
Answer: we added the sentence “All data met these assumptions”, ie the normality and heteroscedasticity assumptions – this statement denotes that all variables were normally distributed.
line 171-174 - I think that the author should accept the consequence of writing data, since he writes the BMI abbreviation, and the fat, fat-free or total body water should be saved as an abbreviation, e.g. FM, FFM, TBW
Answer line 171-174: following the comment of the Reviewer, the abbreviations FM and FFM have been changed throughout the text, in the tables and in the captions.
table 2 should be in a horizontal orientation - it will be more readable
Answer table 2: we agree with the referee. However, a large amount of results does not allow the table to fit within the page margins. Despite this, an example of table 2 in horizontal orientation has been included in the paper.
the description of table 2 should be much shorter, while the current information in the table title moved under the table as a legend. This note also applies to table 3 and 4
Answer: in order to better explain our tables, the description of table 2, 3 and 4 have been shortened and the current information displaced in the notes below the table.
table 3 - I think p = 0.000 is incorrect, I suggest changing to p <0.001 everywhere
Answer table 3: the presentation of p value has been changed.
I would suggest modifying the presentation of the conclusions
Answer: as Reviewer rightly points out, the conclusion section was completely rewritten to better summarize the main achievements and results of our study.
You should change the style of citing from Harvard over Vancouver - non-compliance with the journal's guidelines
Answer: we made a mistake with the journal’s guidelines. All the text has been revised and now the style of citing is Vancouver.
We again thank the Associate Editor and the Reviewer for their thoughtful comments. We hope to have addressed all your concerns.
Reviewer 2 Report
This manuscript in interesting and presents a good structure. However, some details could improve the paper.
Abstract
- Indicate the design of this research and more data of the participants (e.g., average age).
Introduction
- The aim of this research should clearly be described at the end of the introduction.
Methodology
- The authors must explain the research’s design and the selection of the sample (inclusion and exclusion criteria).
- Please, explain if some type of blinding was conducted during collection and during the assessment of this study.
- As well, explain if there were attrition of the sample and why (if yes).
References
- Only 10 references (25%) belong to the last five years. The manuscript must be updated. At least 40% of the references should belong to the last five years.
Author Response
This manuscript in interesting and presents a good structure. However, some details could improve the paper.
We would like to thank the Reviewer for his/her comments that improved the quality of the manuscript. The manuscript was modified according to the suggestion of the Reviewer (see point by point answers below).
Abstract
Indicate the design of this research and more data of the participants (e.g., average age).
Answer: as suggested by the Reviewer, the sentence of the abstract was rewritten in order to better explain our methodological choices.
Introduction
The aim of this research should clearly be described at the end of the introduction.
Answer: we thanks the reviewer for his comment. The aim of the study is now more clearly stated at the end of the introduction.
Methodology
The authors must explain the research’s design and the selection of the sample (inclusion and exclusion criteria).
Answer: The study design is observational. This information was added in the abstract and in the design and methodology section. As rightly suggested by the Reviewer, we included more details in the Materials and Methods section. In the participants session the following sentence was added: “The following inclusion criteria were adopted: i) a minimum of 5 years of previous swimming competitive experience; ii) chronological age between 12 and 19 years old; iii) swimmer able to compete in all the Olimpic swimming strokes and distance; iiii) swimmer that participates in the training for a minimum of 4 session per week in the last 12 weeks.”
Please, explain if some type of blinding was conducted during collection and during the assessment of this study.
Answer: only one blinding was used for the study participants. The following sentence was included in the design and methodology section: “To preserve blinding, the participants were not informed about their individual scores until the end of the study.”
As well, explain if there were attrition of the sample and why (if yes).
Answer: Thank you for the suggestion. The study-related characteristics were completely random and none of the participants drops out the study. Then, the attrition rate of the sample was null. A sentence to better explain this has been added in the design and methodology section.
References
Only 10 references (25%) belong to the last five years. The manuscript must be updated. At least 40% of the references should belong to the last five years.
Answer: as requested by the Reviewer, we updated the references list with more recent papers and deleted some papers not strictly needed. Now 40% of the total references are belong the past 5 years.
We again thank the Associate Editor and the Reviewers for their thoughtful comments. We hope to have addressed all your concerns.
Round 2
Reviewer 1 Report
Thank you for making the correct corrections in the text. In my opinion, the text meets the publication requirements.